# A Novel Enhanced Methodology for Position and Orientation Control of the I-SUPPORT Robot

**DOI:** 10.3390/biomimetics10080502

**Published:** 2025-08-01

**Authors:** Carlos Relaño, Zhiqiang Tang, Cecilia Laschi, Concepción A. Monje

**Affiliations:** 1Systems Engineering and Automation Department, University Carlos III of Madrid, 28911 Madrid, Spain; cmonje@ing.uc3m.es; 2Department of Mechanical Engineering, National University of Singapore, Singapore 117575, Singapore

**Keywords:** soft robotics, fractional-order control, kinematic decoupling, I-SUPPORT, robotic control strategies

## Abstract

This study presents a novel method for controlling the position and orientation of the bioinspired I-SUPPORT soft robot, which represents a relevant advancement in the field of soft robotics. The approach is based on module actuation decoupling and fractional-order control, offering a more advanced and robust control solution. This innovation enhances the versatility of the robot and illustrates the efficacy of fractional-order controllers, which are comparable to current meta-learning-based controllers. The research involves experiments in both vertical and horizontal configurations, addressing tasks ranging from simple orientation to complex interactions, such as gentle rubbing during bathing activities with the robot. These experimental results exemplify the efficacy of the proposed control strategy and provide a foundation for future research in soft robotics control, underscoring its potential for broader applications and further technological advancement.

## 1. Introduction

The flexibility, adaptability, and safety features of soft robots have facilitated the exploration of novel possibilities in diverse fields, including healthcare, manufacturing, and space exploration [1,2,3]. The potential of soft robotics has been recognized in the context of rehabilitation and assistive devices, leading to innovations such as the I-SUPPORT system [4], Figure 1. However, the control of these systems, particularly the precise manipulation and orientation of soft arms, presents considerable challenges due to their inherent compliance and non-linear material properties [1,5]. This study aims to develop an advanced control methodology for the I-SUPPORT system to enhance its precision and robustness, thereby extending its applicability and effectiveness in real-world scenarios.

### State of the Art

The field of soft robotics has garnered significant interest due to its potential to facilitate safer human–robot interactions and adaptability to unpredictable environments. However, despite these advantages, the control of soft robots presents unique challenges, including the management of their infinite degrees of freedom, non-linear dynamics, and the influence of external factors [1]. A variety of control strategies have been investigated, including model-based approaches and bio-inspired methods [6,7]. Nevertheless, achieving an equilibrium between precision and adaptability remains a challenging endeavor.

In rehabilitation and assistive scenarios, seamless control is of paramount importance to ensure user safety and comfort. Systems such as I-SUPPORT are designed to provide autonomy and assistance in personal care tasks, necessitating the implementation of robust control strategies that take into account the user’s specific requirements [2,8]. Moreover, recent reviews highlight that progress toward intelligent soft robots depends on tightly integrated actuation–sensing and on soft, tissue-compliant materials that enable safe, long-term human interaction [9,10].

System identification is a crucial step in the development of precise control models for soft robotics [11]. The incorporation of soft and deformable materials introduces a level of complexity that presents a significant challenge in this regard. A significant challenge in this field is the development of accurate kinematic models and control strategies. The deformation behaviors of these materials are non-linear and unpredictable, resulting in stretching and flexing. This renders traditional kinematic models, which are well suited for rigid bodies, inadequate. In soft robotics, it is imperative to consider these deformations in a comprehensive manner, as they have a significant impact on the robot’s performance and control dynamics. In contrast to conventional robotics, where minor deformations and bending of links are often disregarded as negligible, in soft robotics, they cannot be overlooked. This underscores a fundamental distinction between soft and traditional rigid robotics, emphasizing the distinctive challenges and requirements of controlling systems that exhibit substantial elasticity and flexibility.

The I-SUPPORT robot employs a distinctive hybrid actuation system that integrates the functionalities of McKibben-based actuators and cable-driven mechanisms. This combination exploits the high force-to-weight ratio, flexibility, inherent compliance of McKibben actuators, and the precision and reduced elasticity provided by cable-driven systems. The combination is designed to optimize strength, adaptability, and precise control, particularly in environments where space and maneuverability are constrained.

The accurate control of position and orientation is a crucial aspect of soft robotic arms, and it is therefore essential that these devices are equipped with advanced mechanisms to achieve this goal. Prior research has indicated a preference for pneumatic or hydraulic actuation in soft robotics, given the compatibility of these technologies with the inherent compliance of soft robots [12]. Nevertheless, these actuation strategies are not without shortcomings, including hysteresis and latency in response. Alternatively, shape memory alloys (SMAs) or dielectric elastomers have been investigated as potential actuation strategies, with promising results [13,14]. While these alternatives present promising solutions, they also present challenges, including delayed actuation times and elevated energy demands [15]. This underscores the intricate nature of attaining optimal actuation in soft robotics.

The control strategies for soft robotics are diverse, ranging from adaptive control to reinforcement learning [16,17]. These approaches can offer solutions to the inherent complexities of these systems. However, they often require substantial computational resources or direct human oversight. Therefore, there is an opportunity to develop more streamlined and efficient control strategies.

Meta-learning-based control is a promising alternative in this context, demonstrating notable effectiveness [18,19,20]. This method uses the principle of meta-learning to take advantage of a robot’s previous experience in different environments, allowing it to quickly adapt to new challenges or changes in a new environment. As a data-driven approach, meta-learning-based control capitalizes on accumulated knowledge to adjust models for novel scenarios, prioritizing adaptability and the capacity for generalization. These attributes make it an attractive option for improving the performance and versatility of soft robotic systems.

The application of meta-learning-based control is particularly pronounced in complex, dynamic systems like soft robotics. These systems, defined by flexibility, compliance, and inherent complexity, necessitate real-time control strategies that adapt to evolving conditions and unforeseen environmental interactions. This approach is inherently flexible, enabling rapid adaptation to changes but at the cost of higher computational complexity and the need for extensive datasets. It excels in applications where the environment or system dynamics are non-linear and subject to frequent changes, making traditional modeling challenging.

A comprehensive survey by [21] delves into the field of meta-learning, highlighting its potential to overcome traditional challenges in deep learning, such as data and computation bottlenecks, and enhancing generalization capabilities. Additionally, ref. [22] applies meta-learning for adaptive model predictive control, showcasing its effectiveness in utilizing data from prior tasks to swiftly adapt to new tasks, particularly in the context of autonomous racing and adapting to unseen road conditions, which mirrors the adaptability and rapid learning critical for soft robotics and other dynamic systems.

Machine learning strategies for system identification have shown promise but demand extensive data and computational resources [6,23]. Machine learning algorithms, especially deep learning algorithms, require large amounts of data and significant computing resources, which poses difficulties in real-time applications. In addition, they often lack transparency in their decision-making (“black box” problem), making it difficult to diagnose or predict actions, and they can sometimes react unpredictably [24]. On the other hand, fractional-order proportional integral derivative (FOPID) controllers, generally designed using transfer functions, rely on a mathematical model of the system’s dynamics and incorporate fractional calculus to enhance the control performance and robustness, offering predictability and precision in systems where the dynamics are well understood and can be accurately modeled.

Fractional-order controllers extend the concept of traditional PID controllers by introducing integrators and differentiators of non-integer (fractional) orders, offering more flexibility and potentially better performance in controlling dynamic systems. This type of control has been gaining attention for its robust performance across various disciplines and its ability to handle complex dynamic systems more effectively than conventional PID controllers.

The foundational paper [25] introduces the concept of fractional-order $PI^{\lambda }D^{\mu }$ controllers, which involve fractional-order integrators and differentiators. The paper discusses the use of a new function of Mittag–Leffler type to obtain explicit analytical expressions for the unit-step and unit-impulse response of a linear fractional-order system with a fractional-order controller for both open and closed loops. This work demonstrates the advantages of $PI^{\lambda }D^{\mu }$ controllers over traditional controllers, highlighting their potential for enhancing control system performance. The work in [26] emphasizes the robustness and performance improvements that fractional-order control can offer compared to integer-order control systems.

After a comprehensive examination of the limitations of the meta-learning-based control and the robustness characteristics of the fractional-order control, this work proposes a novel control approach for the I-SUPPORT system. A combination of advanced control techniques based on fractional-order controllers and a kinematic decoupling methodology of the system is proposed in order to improve the performance and robustness of the system and overcome the limitations that have been encountered with machine learning techniques. Furthermore, this approach will use a mathematical model rather than one generated from data, which will allow for a more comprehensive understanding of the system and the relationship between the robot’s modules. Consequently, new degrees of freedom, such as the orientation of the robot’s tip, which has not been previously exploited, will now be defined and controlled. By addressing the aforementioned challenges, this research contributes to the development of a more efficient robotic assistance solution, which, for the first time, allows for the control of both the position and the orientation of the I-SUPPORT robot.

## 2. Materials and Methods

### 2.1. Description of the Platform

This work employs the bioinspired soft robotic platform I-SUPPORT, a serial two-module soft arm designed to assist elderly people in bathing tasks [4], Figure 1. Its features include a structure consisting of adaptive modules that reduce the complexity of active control and a hybrid actuation strategy that combines McKibben-based actuators and tendon-driven mechanisms to achieve desired movements (Figure 2). This combination of technologies ensures effective and safe bathing assistance to the elderly.

It is a lightweight module design weighing 220 g, 17 cm long, and connected by a rigid 3D-printed connector. Dedicated activation lines and Bowden cables enable independent control of the modules and actuators. The manipulator’s overall length is 37 cm.

Each module includes three pairs of McKibben-based actuators and three cables alternately displaced at an angle of $60^{\circ }$ along a circle with a radius of 3 cm. McKibben-based actuation has a triangular arrangement, and cable actuation has an inverted triangle arrangement. While the pneumatic chambers are inflated by compressed air to achieve various movements of the manipulator, the cables are tensioned by servomotors to provide an antagonistic action to the fluidic actuators, resulting in the stiffness of the module when actuated together.

The module has two terminals: “end plate” and “base plate”. These terminals allow quick and direct connections between the two modules and external elements, such as a support frame. In addition, twenty-one rings are placed along the actuator chambers, which maintain the correct working orientation and accommodate the cables.

The pneumatic actuation box is a critical component of the system. It is divided into a pneumatic section with six specialized microactuators that ensure precise manipulation, a robust power supply equipped with safety measures such as fuses to prevent electrical mishaps, and a control system using an Arduino Due that manages the microactuators efficiently.

The six servomotors are enclosed in a specially designed Plexiglas structure. Each can exert a torque of 1.32 Nm. This configuration ensures correct cable alignment and simplifies maintenance tasks such as component replacement.

Unlike other works where I-SUPPORT works only vertically [4], this research will employ both vertical and horizontal configurations. The latter results in a quantum leap of difficulty because of the forces produced by gravity affecting the deformation of the soft sections, especially when handling loads. This modular manipulator is composed of two assembled modules: the first is called the proximal module and is driven solely by a cable-driven system to compensate for gravitational effects; the second is called the distal module, and both are based on the same hybrid actuation (cable and pneumatic).

The cable action allows the module to flex and the end of the module to move in space. Given that the length of the module remains constant, it can be mapped with two degrees of freedom (DoF). The motors have absolute encoders that allow for high movement repeatability at the cost of having a limited range of actuation. Pneumatic actuation would allow the length of the module to be varied, so this actuation could be mapped with two or three DoF if the length is varied as a parameter. However, this makes the control more complex and may decrease the accuracy, so it should be studied for which cases it is suitable for and whether it would be necessary to incorporate this third DoF.

The sensorization of the platform includes the position encoder of the motors for the actuation with cables, sensors to measure the pressure of the pneumatic chambers, and an Intel RealSense depth camera placed in a structure in the upper part oriented towards the arm that allows obtaining information about the position, depth, and RGB. Using color labels on the soft arm makes it possible to detect the XYZ position of an element. The Intel RealSense depth camera used has a resolution of approximately 1 mm, and measurement noise can introduce small fluctuations. However, it is sufficiently accurate for the applications proposed in this work. The distribution of the coordinate axes obtained through the camera can be seen in Figure 3.

### 2.2. Kinematic Model

The I-SUPPORT platform offers multiple performance possibilities due to its diverse range of DoF. At present, a study such as that described in [18] has been able to control the end effector of the soft robot through the application of meta-learning and model-based optimal control. This approach is focused on controlling the X- and Z-positions in space, and it is also capable of controlling the orthogonal force exerted through a force sensor at the distal end of the robot.

#### 2.2.1. Meta-Learning Approach

The methodology followed in [18] hinges on three pivotal components: a target-oriented proactive search, meta-learning with an online adaptation of a data-driven model, and model-based optimal control. The proactive search aims to efficiently gather environment-specific data, reducing unnecessary interactions by employing active learning principles, notably differing from passive or random search methods. The proactive search strategy uses Bayesian optimization to find optimal control actions because Bayesian optimization is an efficient approach to finding optimal values of expensive cost functions. After the data collection phase, meta-learning is applied, which uses accumulated experience to rapidly update a data-driven probabilistic model online. The model is based on the multitask Gaussian process method, which captures the dynamics of the system, including non-linearity and uncertainty. The model is a foundational basis that encapsulates learned patterns and behaviors from previous environments. Meta-learning leverages historical data for rapid model adaptation to new environments, addressing the limitations of previous data-driven methods that struggled with environmental changes.

Nevertheless, this method for controlling the I-SUPPORT soft robot exhibits several limitations that constrain its overall performance and versatility. Primarily, the lack of effective orientation control restricts the robot’s ability to perform tasks that require precise angular positioning. Additionally, this method lacks the capacity to regulate intermediate modules within the robot, thereby reducing its flexibility and responsiveness in dynamic environments. Furthermore, the investigation of the robot’s mobility range has been constrained, leading to an inefficient exploitation of its potential. Finally, this approach relies on large initial movements to achieve desired positions and orientations, which can be inefficient and inaccurate, especially in confined or sensitive operational settings.

This research aims to address these limitations by introducing a comprehensive orientation control, a module decoupling methodology, and a fractional-order control solution, thereby avoiding the large initial movements of the robot and enabling it to perform tasks with high precision and reliability. This represents a substantial advancement in the field of soft robotics, enhancing the functionality and potential applications of the I-SUPPORT soft robot.

#### 2.2.2. The Role of Orientation in Control

Controlling the orientation of the I-SUPPORT soft robot is crucial for several reasons. Firstly, precise orientation control enhances the robot’s ability to perform tasks that require accurate angular positioning. Secondly, by maintaining the desired orientation, the robot can better adapt to external forces and maintain its intended path or position, reducing the risk of errors or accident. This level of control ensures that the robot can interact with objects and surfaces from the correct angles, for example, in bathing tasks such as rubbing a back, improving task efficiency and effectiveness.

In the case of complex robots such as I-SUPPORT, it is not feasible to fully define the position of the soft robot using only the X- and Z-axes without incorporating the orientation as a variable. It is, therefore, imperative that orientation is included in the control process.

In the absence of a clearly defined orientation, cases like in Figure 4 can be found, where for a given same final position for $X_{21}$ and $X_{22}$, two different configurations allow the robot to reach that position but with different orientations. The orientation of the distal module can be used to condition the proximal module positions. Another problem that can be encountered is that there may exist redundant cases, such as the one presented in Figure 5, where for a value $X_{2}$, different solutions are possible.

The following section will present the proposed kinematic solution developed with the objective of controlling both the position and orientation of the I-SUPPORT robot.

#### 2.2.3. Independent Distal Module Actuation Method

Based on an analysis of the previous control method’s limitations and the necessity of orientation control, a new approach is proposed. The Independent Distal Module Actuation (IDMA) method involves independent control of the distal module’s end, allowing for the independent control of both position and orientation.

The IDMA methodology is based on employing the different modules to perform independently the position and orientation control of the I-SUPPORT. In other words, the proximal module is responsible for performing the control of two DoF for positioning, while the distal module is responsible of two DoF in the orientation control. Therefore, the soft robot is controlled by four DoF.

This methodology offers several advantages, as it decouples control in a modular manner, thereby enabling more rapid and precise control of the position and orientation of the distal end. In addition, it is worth mentioning that extending this scheme to add force control at the distal end of the system would be possible, mirroring the meta-learning approach. However, this is beyond the scope of this work and will be considered in future work.

This methodology can be extrapolated to a soft robot with a greater number of modules, but a new angular reference must be added for the intermediate modules. The selection of these configuration parameters will depend on the task to be performed, resulting in a redundant system at the distal end. It should be noted, however, that simultaneous control of both modules is challenging due to their overlapping actions on the same module.

Firstly, an analysis is conducted to determine the impact of the proximal module actuation on the distal module’s end position. A point, designated as $P_{2}\left(X_{2},Z_{2}\right)$, will be commanded within the space for X and Z, which serves as the positional reference for the distal module. Once the control is initiated, the error will be reduced as the distal end approaches the reference point. Under the assumption that the distal module is not subjected to simultaneous actuation or external forces, it can be assumed that the orientation and position of the distal end will remain constant with respect to the coordinate axis at the base of the proximal module, as if it were a fixed coupled element. Consequently, the process of identifying the proximal module is analogous to that of position control, wherein only the proximal module is present.

A similar approach can be employed for the control of orientation using the distal module, but in this case, the reference axis must be changed to the base of the distal module. It should be noted that this robot is lacking an orientation sensor, therefore its identification is based on the Cartesian position. Consequently, the orientation is determined by the position of the preceding module. Therefore, although it would appear that a solution is not possible a priori, a solution can be reached if certain assumptions are made (such as constant curvature flexion) and the error to the reference is redefined as the distance between the end of the module and a straight line in space, as will be explained below.

Accordingly, as previously stated, the initial challenge is to establish a correlation between the Cartesian position XZ and an angular position that is independent of the position and orientation of the upstream module. It can be assumed that if the drive of the proximal module remains constant (i.e., the module does not move), the base of the distal module will remain in a fixed position. Conversely, it is also possible to assume that, in the absence of external forces, the position and orientation of the proximal module remain unaffected by the actions occurring in the distal module. With this in mind, it is possible to represent a straight line passing through the base and the end of the distal module. These XZ spatial coordinates are obtained from the depth chamber, and the orientation of this line will indicate the orientation of the end of the proximal module.

The next challenge arises due to the orientation of the proximal module. When the proximal module’s end undergoes a change in orientation for positioning in space, it becomes evident that the XZ axes are no longer sufficient for the positional control of the module. Instead, the XYZ axes must be employed. This results in a significantly more complex system, as the identified system undergoes a change due to the necessity of controlling three DoF with an initial decoupling designed for two DoF. Furthermore, precise trigonometry is required to ensure that the commanded XYZ position is achievable.

Thus, to avoid utilizing that more complex system with XYZ, the proposed solution to this problem is to create a new reference system situated at the base of the distal module. Rather than developing a control that minimizes the error between the distal module’s end and a point, a control is created that minimizes the error between the distal module’s end and a straight reference line.

This methodology enables the system to be simplified, as illustrated in Figure 6 and Figure 7. The use of a straight line instead of a point allows for the elimination of the additional DoF without compromising the desired angle. For each DoF, a reference line is calculated by performing a displacement $P_{2}^{\prime }\left(X_{2}^{\prime },Z_{2}^{\prime }\right)$ in the X’- or Z’-axis reference with respect to the line joining the proximal and distal ends. This results in a reference line that is parallel and therefore has the same slope. The X’- and Z’-axes serve as the new reference axes for the values of $P_{2}^{\prime }$.

The straight line generated from the displacement input value is independent of the orientation and position of the proximal module. The movement made by the distal module for a given value would always be the same type of flexing towards the straight line, and that flexing would end up corresponding to a given orientation of the end.

In the case of the distal module, the error provided to the controller is the distance between the distal end position value and the reference line. This can be calculated through the distance equation between a point P ($p_{1},p_{2}$) and a straight line r (Ax+By+C=0) :
(1)$$d\left(P,r\right)=\frac{|A\cdot p_{1}+B\cdot p_{2}+C|}{\sqrt{A^{2}+B^{2}}}$$

It should be noted that the case of the proximal module can be considered a trivial case of this implementation with respect to the distance of a straight line. In this case, neither the orientation nor the position of the base is changed; however, the reference axis of origin is used so that the error is calculated directly.

As a result, this methodology enables the first independent control of the orientation and position acting on the distal and proximal modules, respectively. The ability to control both independently and modularly offers significant advantages over a system where both are dependent. The methodology employed at the distal end enables the straightforward and consistent attainment of the same orientation value upon entry of the identical reference value, obviating the necessity for more intricate XYZ references.

Nevertheless, this methodology focuses on the fact that each module will be responsible for two DoF, despite the actuation system consisting of three cables and three pneumatic actuators. It is, therefore, necessary to identify a method of decoupling the X- and Z-axes from the actuation system, as will be discussed in the following section.

#### 2.2.4. Actuation

Following the analysis of the IDMA method, a question arises concerning the actuation system to be used for each module. Given that each module is equipped with both a cable actuation and a McKibben-based actuation, as previously stated in the I-SUPPORT description, it is necessary to determine which of these systems should be employed for each module.

It is possible to achieve an actuation in which both cables and McKibben actuators are controlled simultaneously. Nevertheless, this simultaneous actuation has been observed to introduce complexity into the system, which has led to the decision to actuate each module with a single type of actuation.

The application of cable control in the proximal module is characterized by a high degree of accuracy. However, in the distal module, there is a limitation in relying on the cables to exert sufficient and repeatable tension due to the influence of the proximal module’s curvature on the cable tension.

Conversely, pneumatic actuation seems an appropriate choice for the distal module, given that the pressure exerted is independent of the curvature of the preceding module. Nevertheless, the control of this system is more complex and results in slight increases in vibration.

Consequently, it was ultimately determined that cable actuation will be employed in the proximal module, while McKibben-based actuation will be utilized in the distal module.

### 2.3. Model Identification

As discussed previously, each module has six actuators, that is, three actuators per actuation type, as shown previously in Figure 8. The combination of how they are used is responsible for the positioning and orientation of the module.

The system is defined as a multi-input, multi-output (MIMO) system. However, it is possible to decouple the three actuators for each actuation type with respect to the two flexing angles aligned with the X- and Z-axes. The advantage of decoupling is that the MIMO model can be transformed into a series of single-input, single-output (SISO) systems, which are easier to identify. It is essential to note that, by performing the decoupling, the information about the action of one system on the other may be lost. However, a previous work [27] has demonstrated that the coupling influence is minimal for the I-SUPPORT case, which justifies the decoupling methodology proposed next.

#### 2.3.1. Actuation Decoupling

This process where the system is simplified is of vital importance since it allows to intuitively separate the movements of each module in the two main axes for both actuation methods (cable and pneumatic), which has not been done before.

#### 2.3.2. Cable Decoupling

A new decoupling methodology is developed from that described in [28], in which the X and Z displacements are decoupled from cable length variation through the position of the motors $M_{i}$, as shown in Figure 8. This approach leads to the derivation of the following equations: (2)$$M_{1}=\frac{\alpha_{m}}{1.5}+\omega_{o}$$(3)$$M_{2}=\frac{\beta_{m}}{1.732}-\frac{\alpha_{m}}{3}+\omega_{o}$$(4)$$M_{3}=-\frac{\beta_{m}}{1.732}-\frac{\alpha_{m}}{3}+\omega_{o}$$
where $M_{i}$ is the motor position for the cable with constant curvature and $M_{1}+M_{2}+M_{3}=3\cdot \omega_{o}$, where $\omega_{o}$ refers to the offset value of the motors. I-SUPPORT motors use absolute encoders, and for this case, the rest position is $\omega_{o}=450$. The values of $\alpha_{m}$ and $\beta_{m}$ refer to the displacements in Z and X, respectively, and will be the input values for the system identification. Thus, the corresponding motor position is obtained by assigning a value of $\alpha_{m}$ and $\beta_{m}$. If a value of $\beta_{m}=0$ is fixed and the value of $\alpha_{m}$ is varied, the robot will move in the Z-axis; while if the value of $\alpha_{m}=0$ is fixed and the value of $\beta_{m}$ is varied, the robot will move along the X-axis. In this case, as can be seen from Equation (Equation 2) or on the soft robot itself, $M_{1}$ would not move because it does not affect the motion along the X-axis.

#### 2.3.3. Pneumatic Decoupling

The decoupling of the pneumatic actuation is similar, although some aspects must be taken into account. When performing a pneumatic actuation $P_{i}$, as shown in Figure 8, the module flexes. However, the amount of pressure applied also affects the length of the module, resulting in an elongation. In other words, if the three actuators are actuated with the same pressure, there will be no bending but rather an elongation, which represents a new variable that must be decoupled. This is possible to be performed following the work [27], where a decoupling is conducted from the values of $\alpha_{p}$, $\beta_{p}$, and a third value that is related to the change in length, $l_{p}$. The following equations are derived:(5)$$P_{1}=\alpha_{p}+\frac{P_{2}+P_{3}}{2}$$(6)$$P_{2}=\beta_{p}+P_{3}$$(7)$$P_{3}=l_{p}-min\left(\alpha_{p}+\frac{\beta_{p}}{2},\beta_{p},0\right)$$
where $P_{i}$ is the pressure in the pneumatic actuator. In this case, the rest position corresponds to zero pressure in all the actuators, so the offset value is $\omega_{o}=0$. The values of $\alpha_{p}$ and $\beta_{p}$ are related to the displacements in Z and X, respectively. However, it is essential to clarify that they do not refer to the same values as in the case of cable decoupling ($\alpha_{m}$ and $\beta_{m}$). In addition, due to the inverted triangular distribution of the pneumatic actuation, the displacements will have the opposite sign with respect to the cable actuation. Finally, one can consider $l_{p}=0$ for simplicity, meaning that the decoupling prevents the modulus length variation by providing pressure values that would not affect the elongation.

#### 2.3.4. Module Identification

The experiments will be performed in two different configurations of the I-SUPPORT robot: vertical and horizontal. These configurations will change the dynamics of the system, so a specific identification will be made for each of them.

Firstly, the study will be conducted with the soft robot in a vertical arrangement, with the origin at the top. In this case, the distal module’s end position will be identified.

Secondly, the analysis will be conducted in a horizontal configuration. This configuration is more challenging than the previous case because it will produce greater moment forces due to the combined effects of gravitational action and distance to the origin. In this instance, the objective is to identify the position and orientation of the distal module’s end.

The identification of each module is performed through two transfer functions $G_{\alpha }\left(s\right)$ and $G_{\beta }\left(s\right)$, which represent the relationship between an α input and the displacement of the robot tip in the Z-axis and the relationship between a β input and the displacement in the X-axis, respectively. These functions model the dynamics of the system and facilitate the latter design of the controllers of the robot.

The X- and Z-data of each module were obtained using the Intel RealSense depth camera by placing a colored tape so that, by segmenting the RGB values, the value of the position of the distal end in millimeters can be obtained. For the identification method, the parameters are initialized using the Instrumental Variable (IV) method, and their updates are performed using a non-linear least squares search approach [29].

#### 2.3.5. Vertical Configuration Identification

In the vertical configuration, the effect of cable actuation of the proximal module on the end position of the robot will be identified.

The aforementioned decoupling process allows us to obtain the required data for the identification of the Z- and X-movements, resulting in the two transfer functions $G_{\alpha }\left(s\right)$ and $G_{\beta }\left(s\right)$ of Equations (Equation 8) and (Equation 9), respectively. The identification process involves defining specific parameters, such as the number of poles, zeros, and delay of the transfer functions. In this case, the parameters that best described the system were two poles, no zeros, and no delay.(8)$$G_{\alpha }\left(s\right)=\frac{-7.115}{s^{2}+4.347s+6.414}$$(9)$$G_{\beta }\left(s\right)=\frac{6.923}{s^{2}+4.134s+6.623}$$

Note that being in an opposite arrangement to the cable distribution results in a transfer function with opposite gain sign to the proximal module.

#### 2.3.6. Horizontal Configuration Identification

In the case of the horizontal configuration, the identification of the movements in the X- and Z-axes for the proximal module and in the X’- and Z’-axes of the distal module will be carried out following the IDMA methodology. Thus, four identifications are necessary, all of them related to the end position of the distal module.

The identifications of the proximal module are denoted by sub-index 1 (Equations (Equation 10) and (Equation 11)), and those of the distal module are denoted by sub-index 2 (Equations (Equation 12) and (Equation 13)). As in the vertical configuration, α represents the motion in the Z-axis and β in the X-axis.

After performing the identification, the transfer function parameters that best defined the system were two poles, no zero, and no delay, except for $G_{\alpha 2}\left(s\right)$, with three poles, raising the following transfer functions:(10)$$G_{\alpha 1}\left(s\right)=\frac{-7.863}{s^{2}+7.044s+10.23}$$(11)$$G_{\beta 1}\left(s\right)=\frac{9.542}{s^{2}+6.614s+11.37}$$(12)$$G_{\alpha 2}\left(s\right)=\frac{595.6}{s^{3}+12.08s^{2}+178.4s+1295}$$(13)$$G_{\beta 2}\left(s\right)=\frac{-132.2}{s^{2}+17.15s+119.8}$$

Although pneumatic actuators typically exhibit delays, in our setup, these were experimentally observed to be negligible due to the small dimensions of the robot and the limited air volumes of its chambers, which remain partially inflated during operation. Consequently, the identified transfer functions were modeled without delay; an assumption supported by the experimental results showing stable and accurate tracking without noticeable latency effects. Delay modeling may, however, be required in larger or faster pneumatic systems.

## 3. Fractional-Order Controller Design

After introducing and applying the novel modeling methodology proposed before, the design of the fractional-order controllers will now be introduced.

### 3.1. Tuning Method

This section describes the tuning method followed for the design of a fractional-order proportional integral (FOPI) controller. The FOPI controller was chosen in this case for its inherent advantages in addressing the unique challenges presented by the I-SUPPORT robot. The integral component of the FOPI controller provides robustness against steady-state errors and parameter variations prevalent in soft robots’ inherently flexible and deformable structures. The derivative term is frequently employed to enhance transient response and stability, but it can also amplify noise and cause instability in systems with measurement noise or high-frequency dynamics. In soft robotics, where sensors may be susceptible to noise due to the robot’s flexible nature, it is preferable to avoid the derivative term as it can result in an unstable robot performance.

Incorporating fractional-order elements within the FOPI controller provides enhanced tuning flexibility. This feature is crucial in achieving precise control over the robot’s complex dynamics, eliminating the need for a derivative component and streamlining implementation. Consequently, the FOPI controller is an excellent option for ensuring performance stability and reliability in real-time applications. This perfectly aligns with the critical requirements for effective control in soft robotics.

As for the design specifications, constraints are imposed on two critical design parameters of the open-loop control system: the phase margin ($\phi_{m}$) and the gain crossover frequency ($\omega_{gc}$). These specifications have been chosen for their ability to influence the system behavior. The phase margin allows us to adjust the damping ratio, while the gain crossover frequency affects the response speed and stability of the system. The controllers were designed to work at a frequency of 10 Hz, since the camera needs 0.1 s to process the XYZ position information. This feature, in addition to the consideration of a reduced overshoot in order to limit the negative effects of the bumping of the elastic modules, lead to the selection of a gain crossover frequency $\omega_{gc}=1$ rad/s and a phase margin $\phi_{m}=70^{\circ }$.

#### Iso-m Tuning Method

Using the control specifications described before, the corresponding FOPI controllers are now tuned using the iso-m tuning method described in [30].

Load variations have a major effect on the nominal gain of the I-SUPPORT system, underscoring the need for a controller that can robustly adapt to such variations. This method takes advantage of the fractional order’s ability to modify the slope of the open loop phase curve, ensuring a flat phase profile. As a result, the phase remains unchanged across different crossover frequencies and system gains, and so does the phase margin. This constraint ensures that gain variations do not result in overshoot changes, which is critical to maintaining system robustness. It also provides stability over the full gain range, which is critical for reducing unwanted deflection when handling high loads.

The necessary steps to tune the controllers following the iso-m method are described below.

First, determine the system phase $\phi_{s}$ and phase slope $m_{s}$ at the gain crossover frequency, $\omega_{gc}$. These parameters define the system’s response characteristics at this frequency. The design of the controller should aim to compensate for the phase slope of the system. To achieve this, the controller’s phase slope should be equal in magnitude but opposite to that of the system. Additionally, the controller must provide a specific phase contribution at $\omega_{cg}$ to meet the desired phase margin specifications, $\phi_{m}$. This phase contribution can be calculated by adjusting the initial phase of the system to account for the desired phase margin and then subtracting 180 degrees to conform to control theory conventions.

Once this is done, it is necessary to determine the fractional order of the controller, represented by λ. This can be done by utilizing the phase slope correction and the required phase contribution identified previously. After determining the fractional order, the time constant represented by $\tau_{a}$ is calculated. This is essential for configuring the controller’s time-domain response. More details on calculating λ and $\tau_{a}$ are given in [30].

Finally, the controller gain, *k*, is calculated using the fractional order λ and the time constant $\tau_{a}$. The gain is crucial for achieving the desired amplitude response characteristics of the control system and concluding the tuning process by setting the controller parameters, which include the proportional gain ($k_{p}$), the gain associated with the fractional action ($k_{a}$), and the fractional order (λ), as given in Equation (Equation 14). Note that positive values of λ will be obtained as a derivative action, and negative values as an integral action.(14)$$C\left(s\right)=k\left(1+\tau_{a}s^{\lambda }\right)=k_{p}+k_{a}s^{\lambda }$$

### 3.2. Resulting Controllers

Next, the controllers obtained after applying the iso-m tuning method considering the transfer functions previously identified are presented.

#### 3.2.1. Vertical Configuration Controllers

For the vertical configuration of the soft robot, the resulting controllers are those in Equations (Equation 15) and (Equation 16) for the position control in the Z- and X-axes, respectively.(15)$$C_{\alpha }\left(s\right)=0.5767+\frac{0.9583}{s^{1.18}}$$(16)$$C_{\beta }\left(s\right)=0.5588+\frac{1.0032}{s^{1.18}}$$

#### 3.2.2. Horizontal Configuration Controllers

Similarly, for the horizontal configuration of the soft robot, the resulting controllers are those in Equations (Equation 17) and (Equation 18) for the position control in the Z- and X-axes, respectively.(17)$$C_{\alpha 1}\left(s\right)=0.771+\frac{1.4420}{s^{1.15}}$$(18)$$C_{\beta 1}\left(s\right)=0.5965+\frac{1.2927}{s^{1.16}}$$

Meanwhile, the resulting orientation controllers in the Z- and X-axes are given by Equations (Equation 19) and (Equation 20), respectively.(19)$$C_{\alpha 2}\left(s\right)=0.2313+\frac{2.2222}{s^{1.2}}$$(20)$$C_{\beta 2}\left(s\right)=0.1021+\frac{0.9290}{s^{1.2}}$$

The Bode diagram of Figure 9 shows how the desired control specifications are achieved for the case of controller $C_{\beta 2}\left(s\right)$, together with the robustness constraint provided by the flat phase curve around the gain crossover frequency. Similar results are obtained for the rest of the controllers.

This robustness is also demonstrated in Figure 10, corresponding to the closed loop time responses of the system for varying gains between 0.6 and 1.4. It can be observed how the overshoot of the responses keeps constant despite this gain disturbance. This specification would not be achievable in general terms for the case of integer-order PID controllers.

There are different methodologies for applying the controllers; this work used the same method described in [28], where the fractional-order controller is approximated to a sixth-order transfer function.

## 4. Results

Fractional-order controllers have been successfully designed for the I-SUPPORT in both configurations. Experiments will be conducted to control the end effector position of the soft robot in a vertical position. The results of these experiments will be compared with those obtained using the meta-learning approach. A series of experiments will also be performed with the robot in a horizontal configuration to test its position and orientation control capabilities for various tasks. It is important to note that a direct comparison with the other methodology cannot be made in this last experiment due to the lack of orientation capabilities.

### 4.1. Vertical Configuration

This section will discuss the experiments performed with the I-SUPPORT robot in a vertical configuration. These positioning-only experiments are intended to show that positioning using decoupling and fractional-order control achieves similar performance to meta-learning.

#### 4.1.1. Experimental Results Using Different Masses

Once the controllers from Section 3.2.1 were obtained, a series of experiments were performed to analyze the behavior of the robot for different loads and distributions, as shown in Figure 11a.

For the X-axis experiments, different reference values $Ref_{x}$ of 78 and −30 (Figure 11b,c) were commanded, and for the Z-axis experiments, the reference values for $Ref_{z}$ were 80 and −40 (Figure 11d,e). For each reference, six experiments were performed, in which masses varying from 0, 10, 20, 30, 50, and 70 g were added at the distal end. Additionally, an experiment was performed in which masses of 20 g were added to the lateral sides of the distal and proximal modules in an unbalanced distribution (Figure 11a). This set of experiments allows for assessing the behavior of the robot under disturbance conditions similar to those that could be encountered during bathing tasks where the robot had to deal with water and a sponge at its tip.

In the experiments, it was observed that I-SUPPORT’s resting position was not zero. This was due to the fact that the robot exhibited a slight misalignment caused by the curvature of its modules in the absence of any action on its parts. However, this initial curvature does not affect control accuracy, since the proposed control strategy regulates motion relative to the resting configuration, and the fractional-order controllers robustly handle this offset without requiring explicit calibration or compensation. Additionally, the responses demonstrated a robust performance of the system before load changes, maintaining a constant overshoot and a speed coherent with the previously defined control specifications.

#### 4.1.2. Comparison of FOPI and Meta-Learning Approaches

To further analyze the controllers’ performance, a series of reference pairs [$Ref_{x}$, $Ref_{z}$] are commanded and the responses of the meta-learning-based control (the current state-of-the-art I-SUPPORT control) and the developed fractional-order controllers are compared. Note that for these experiments, in the case of the fractional-order control, both controllers $C_{\beta }\left(s\right)$ and $C_{\alpha }\left(s\right)$ will act simultaneously to track the combined XZ references. The following position pattern will be defined, commanding each trajectory point every 10 s: [$Ref_{x}$, $Ref_{z}$] = [25, −40], [−25, 18], [25, 80], and [78, 18].

Figure 12 (a–c for the X-axis reference component and c–e for the Z-axis one) shows the performance of the robot for both FOPI and meta-learning approaches under no load, 50 g tip mass, and 20 g side masses attached to the modules.

The experiments yielded the following conclusions. Firstly, despite the fact that both control solutions guaranteed the stability of the system for all load conditions, the meta-learning approach initiated with a few large robots’ movements before reaching the first reference point. This was due to the objective of acquiring environment-specific data, which enabled the optimization of the model. This may be regarded as a disadvantage, as this movement may be suboptimal in specific task contexts. Conversely, it was observed that once the system had learned, it reached the reference faster. For the FOPI case, the system exhibited significantly reduced oscillations, which can be attributed to the controllers’ design for slow and robust dynamics. A more comprehensive evaluation of the FOPI approach is presented next, with a focus on the results obtained in the horizontal configuration.

Table 1 summarizes the main differences between the previously applied meta-learning-based control and the proposed strategy in terms of control structure, orientation handling, initial movement behavior, and adaptability.

### 4.2. Horizontal Configuration

Three different tests were performed to investigate the limitations of both the robot and the FOPI controller in the horizontal configuration: (1) position control, (2) orientation control, and (3) task control. The performance of the robot with different loads was also studied. Note that the meta-learning approach is not evaluated here since it does not provide orientation control.

#### 4.2.1. Position Control

This test aimed at positioning the distal end by actuating the proximal module using controllers $C_{\beta 1}\left(s\right)$ and $C_{\alpha 1}\left(s\right)$.

The robot movement started from the rest position, whose Z component was 320 mm due to the Z-axis orientation in the horizontal configuration. First, a reference pattern [$Ref_{x1},Ref_{z1}$] = [0, 180], [0, 250] was defined, commanding each reference point every 10 s. A raising and lowering movement of the distal end was intended with this experiment. The results are shown in Figure 13, where it is observed how the X-axis movement kept around the 0 component (a) while the Z-axis movement tracked the corresponding Z component (b).

Then, holding the last Z-position, a new reference pattern was commanded for the X-axis as follows: [$Ref_{x1},Ref_{z1}$] = [50, 250] and [0, 250], starting at time 40 s. Finally, a last reference point [30, 200] was commanded in order to change both trajectory components simultaneously.

A similar experiment was then performed with a 50 g mass attached to the tip of the robot. Due to the deformation caused by the mass in the two modules, the Z-components of the trajectory were slightly modified to match the robot’s range of motion, as shown in Figure 13c,d. The results showed that despite the effect of gravity on the robot, which is more significant in the presence of masses, its performance remained robust.

#### 4.2.2. Orientation Control

For the orientation test, controllers $C_{\beta 2}\left(s\right)$ and $C_{\alpha 2}\left(s\right)$ were implemented to command the orientation of the distal module. In these experiments, the distal module was first placed at a given position using the proximal module, and then the control actions were performed only on the distal module to achieve the desired orientation.

The results are shown in Figure 14a,b on the transposed X’- and Z’-axes. These axes are where the orientation control is performed, as explained previously, corresponding to a pitch rotation in Z and a yaw rotation in X, respectively. First, the following reference pattern was defined: [$Ref_{x2},Ref_{z2}$] = [−30, 0], [−15, 0], commanding each reference point every 10 s. Then, at time 20 s, a new profile was commanded: [$Ref_{x2},Ref_{z2}$] = [0, −30], [0, −15]. Finally, a reference point [−20, −20] was commanded in order to change both orientation components simultaneously.

Next, a similar experiment was performed where a 50 g mass was attached to the robot’s tip. Similarly to the position test, the orientation components were slightly modified to adapt to the robot’s range of motion in the presence of mass, resulting in the following pattern: [$Ref_{x2},Ref_{z2}$] = [−20, 0], [−15, 0], [0, −10], [0, 0], [−10, −10]. The results are shown in Figure 14c,d.

It must be remarked that these tests represent a challenging scenario, since the robot is subject to gravity and to a significant load variation (in comparison with its reduced weight). Furthermore, the pneumatic actuation system of the distal module, in comparison with the motor actuation system of the proximal module, introduces greater oscillations into the control loop. Despite all these control challenges, the performance results of Figure 14 demonstrate that the robot could robustly face the load disturbances introduced in the system, keeping a constrained overshoot and tracking the commanded trajectories in all cases.

#### 4.2.3. Task Control

This task aims to replicate the action of rubbing a sponge on a person’s back. It is worth noting that this is a novel experiment on this platform, since previous approaches, like the meta-learning-based control, do not consider the orientation of the robot’s tip, which makes it more difficult to perform the task properly.

The experiment comprised three phases: a first positioning phase ①, an orientation phase ②, and a rubbing phase ③.

The first positioning phase brought the tip of the I-SUPPORT robot closer to the back where the rubbing was to be performed, which took place between seconds 0 and 15. The reference [$Ref_{x1},Ref_{z1}$] = [−30, 225] was commanded to the position controllers. Figure 15a,d ① shows how the positioning is correctly performed in both axes, X and Z.

Next, the soft robot’s tip was oriented to accommodate to the person’s back, which took place between seconds 15 and 30 by the commanding of the reference [$Ref_{x2},Ref_{z2}$] = [30, 0] to the orientation controllers. In this case, the distal module was flexing laterally. It can be observed in Figure 15b ② how the transposed X’ reference was reached using the kinematic methodology proposed, while Figure 15e shows how the transposed Z’ component was kept at 0 value; that is, the robot just bent in the X’-axis. Figure 15a also shows how the displacement caused by this bending was performed during phase ②.

Finally, once the robot’s tip was correctly posed in position and orientation, the rubbing task was commanded following a sinusoidal position pattern that started at second 30, as follows: [$Ref_{x1},Ref_{z1}$] = [−30, 225 + sin(ϕ)25], where ϕ changes 0.1 radians every 0.1 s. Figure 15d ③ shows how the sinusoidal motion in the Z-axis was performed (b) by the distal module, while the positioning in the X-axis was kept constant (a). This performance is in total agreement with that of the proximal module (responsible for the positioning) represented in Figure 15c,f ③.

It is worth noting that, while the constant curvature assumption is widely adopted for its modeling simplicity, it is not strictly satisfied in real conditions due to gravity and load-dependent deformations. In this work, it is considered only as an initial approximation, with a robust fractional-order control strategy compensating for deviations from ideal curvature. Experimental validation in both vertical and horizontal configurations demonstrates accurate position and orientation tracking despite these modeling limitations.

Through all these results, it can be concluded that the task control was performed correctly. Although some slight tracking errors were identified, mainly due to the sensitivity of the actuation systems, they did not compromise the robot’s performance and could be considered negligible in a rubbing scenario like this. The explicit orientation control strategy improved endpoint stability and trajectory consistency, reducing transient oscillations and enabling smoother contact interaction during the rubbing task. This resulted in more repeatable performance compared to previous approaches without orientation regulation and expands the potential of the I-SUPPORT platform to tasks requiring precise alignment with surfaces or instruments.

## 5. Conclusions and Future Works

This work represents significant progress in improving the control capabilities of the I-SUPPORT robot. The position and orientation control of its distal end in the vertical and horizontal configurations has been successfully implemented through the development of the IDMA methodology and the kinematic decoupling proposed. These two approaches together have allowed the implementation of a fractional-order control technique which has provided performance robustness to the system over load disturbances.

The comparison of this approach with the meta-learning-based control has demonstrated the value of this novel solution and has yielded promising results leading to new challenging research issues. This way, future work will focus on refining the effectiveness of fractional-order control in mass handling through adaptive control strategies, similar to how the meta-learning control method performs. In addition, exploring the implementation of force control into the proposed control architecture promises to expand the robot’s capabilities. All this will contribute to improving the performance and adaptability of the robot for a wider variety of tasks in collaboration with humans.

Furthermore, the modular nature of the proposed IDMA control framework allows its extension to I-SUPPORT configurations with additional modules or scaled-up geometries. While the decoupling strategy remains valid, increasing the number of modules or their size amplifies kinematic complexity and actuator loads, requiring re-tuning of fractional-order parameters and potentially more powerful actuation and improved sensing to maintain the same performance.

## Figures and Tables

**Figure 1 biomimetics-10-00502-f001:**
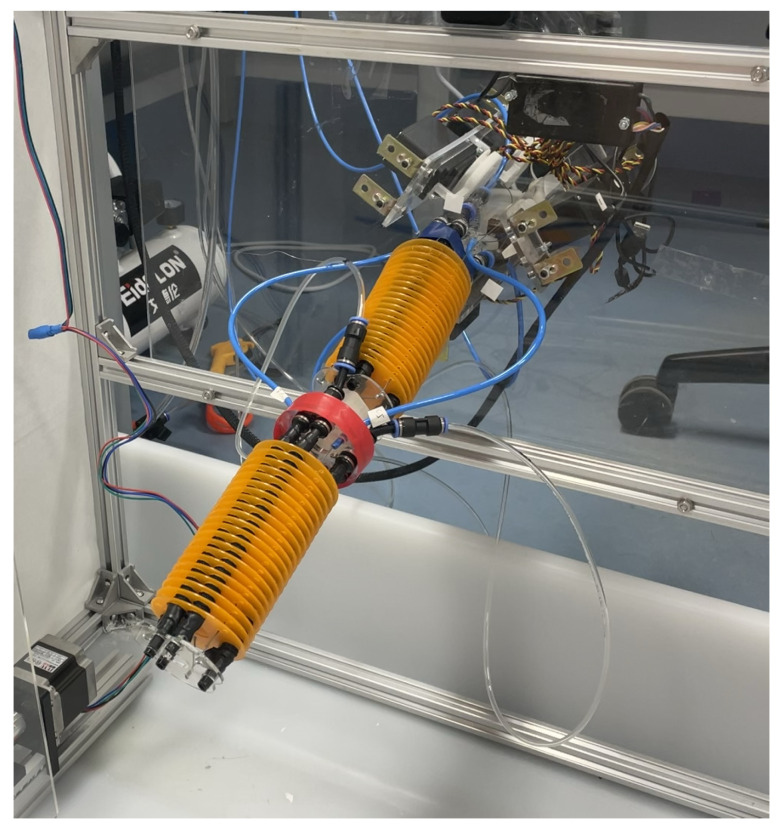
The I-SUPPORT system extended in a horizontal configuration.

**Figure 2 biomimetics-10-00502-f002:**
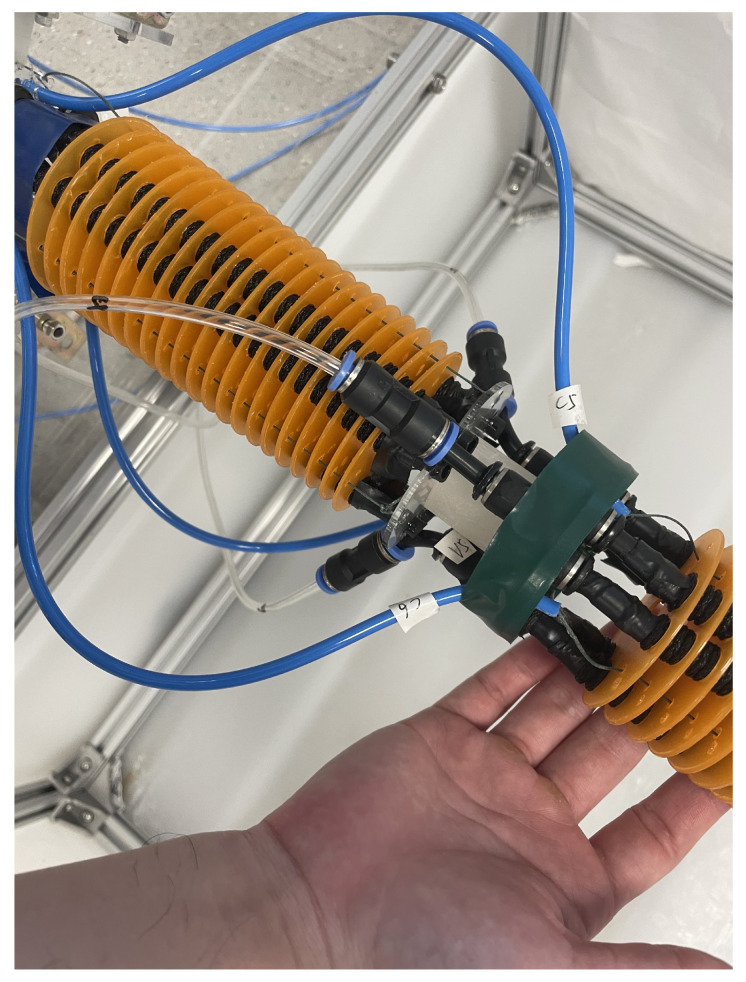
Details of the action on the distal module. It is observed where the cables (blue tubes) and the pneumatic pressure (transparent tubes) are introduced.

**Figure 3 biomimetics-10-00502-f003:**
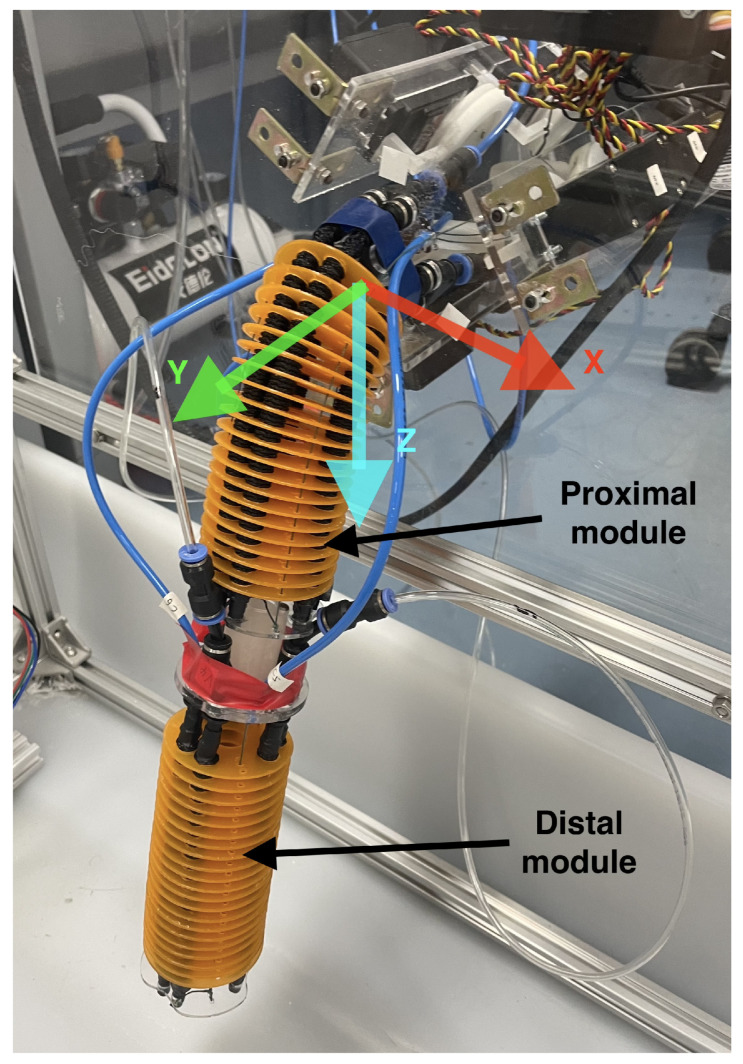
Details of the distribution of the proximal and distal modules. The representation of the axes according to the origin of I-SUPPORT is presented.

**Figure 4 biomimetics-10-00502-f004:**
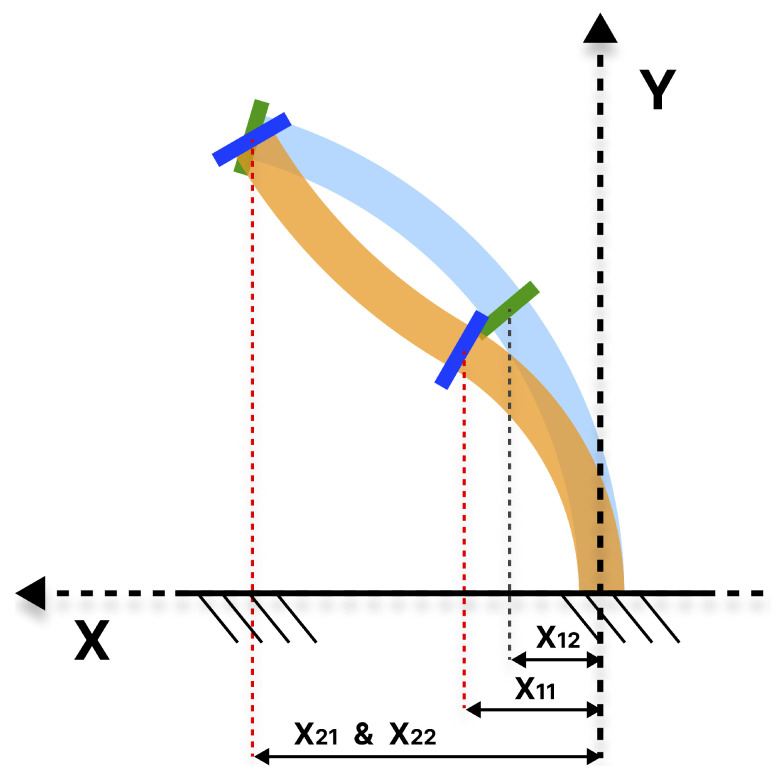
Representation in the XY plane showing two different arms reaching the same final position (orange $X_{21}$ and blue $X_{22}$). However, they have different final orientation values for the proximal module (orange $X_{11}$ and blue $X_{12}$).

**Figure 5 biomimetics-10-00502-f005:**
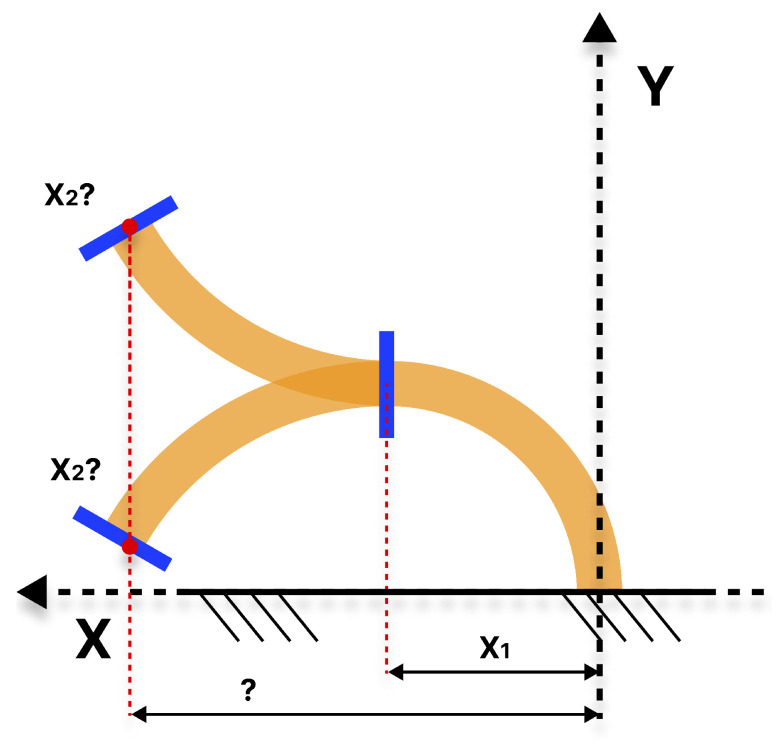
Representation in the XY plane of the soft arm where it is not possible to determine the final position only through the position value of each module. Different solutions exist for the same $X_{2}$ value when the orientation is not conditioned.

**Figure 6 biomimetics-10-00502-f006:**
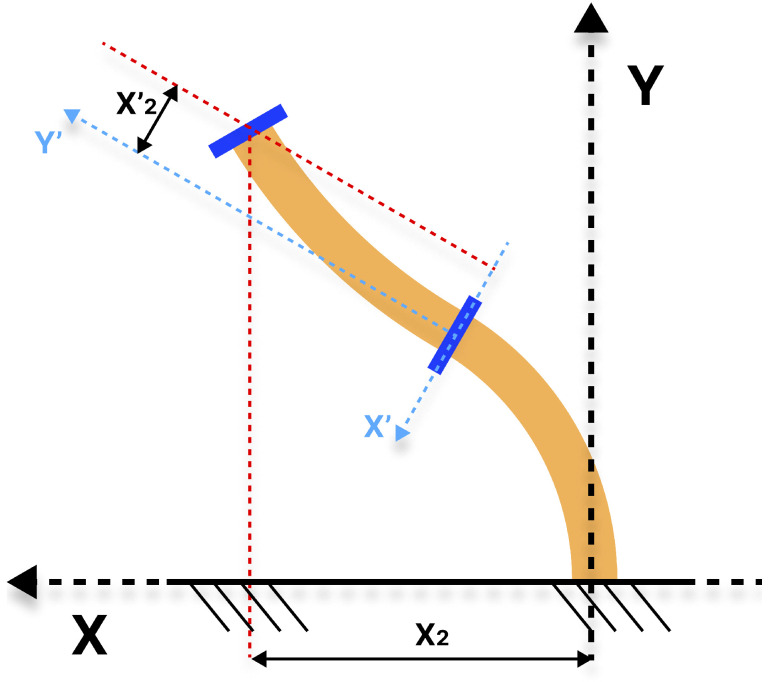
Representation in the XY plane of the $X_{2}$ and $X_{2}^{\prime }$ positions assigned to the soft robot.

**Figure 7 biomimetics-10-00502-f007:**
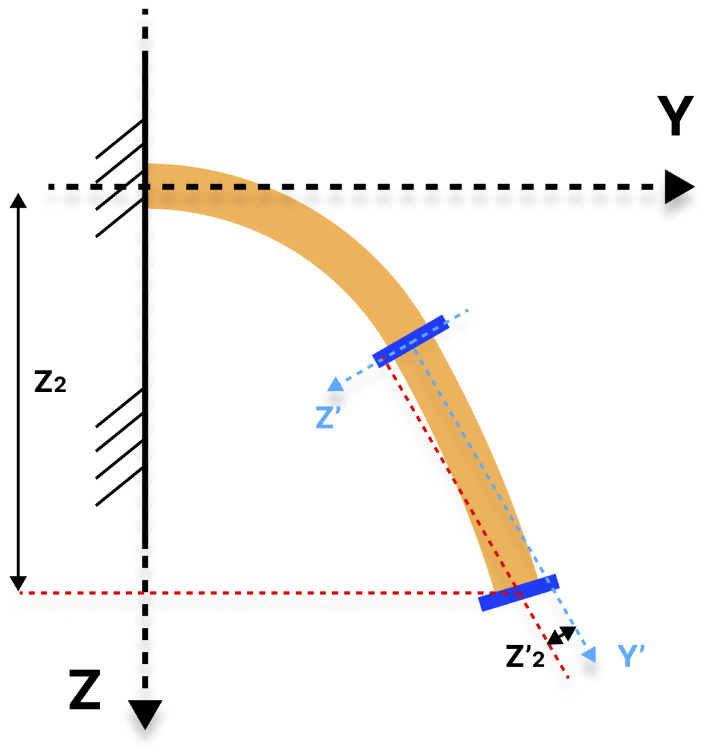
Representation in the YZ plane of the $Z_{2}$ and $Z_{2}^{\prime }$ positions assigned to the soft robot.

**Figure 8 biomimetics-10-00502-f008:**
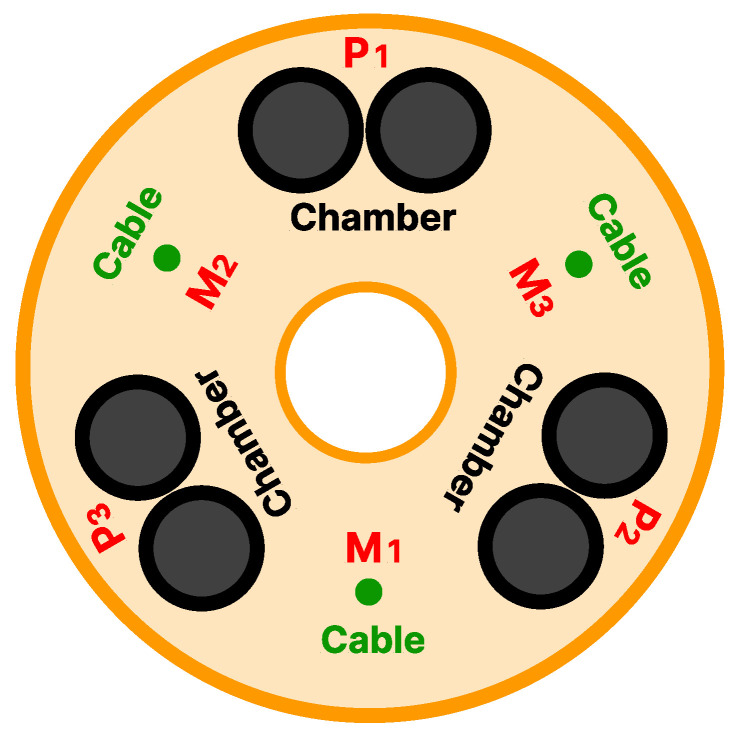
Segment representation of I-SUPPORT. The distribution of the cables actuated by motors ($M_{i}$) and the pneumatically actuated chambers ($P_{i}$) can be observed.

**Figure 9 biomimetics-10-00502-f009:**
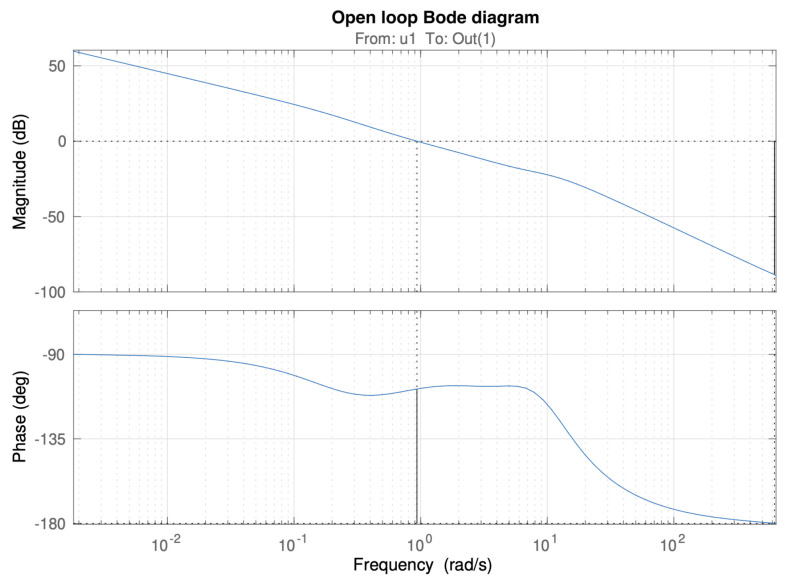
Bode diagram of the open loop system with controller $C_{\beta 2}\left(s\right)$.

**Figure 10 biomimetics-10-00502-f010:**
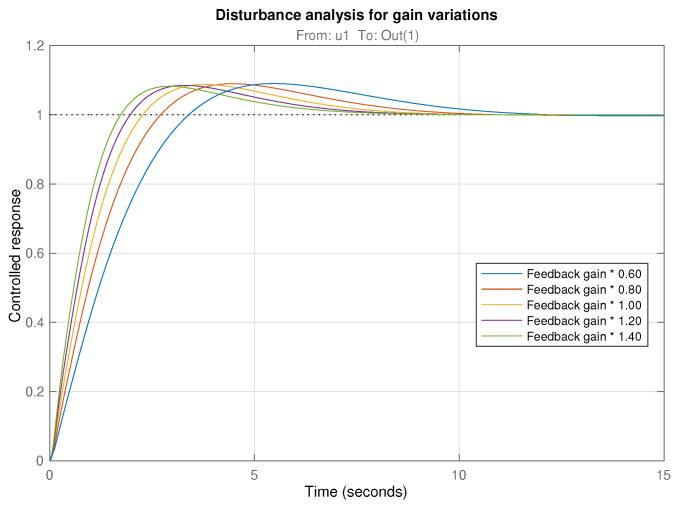
Time responses of the closed loop system with controller $C_{\beta 2}\left(s\right)$ for varying gains from 0.6 to 1.4.

**Figure 11 biomimetics-10-00502-f011:**
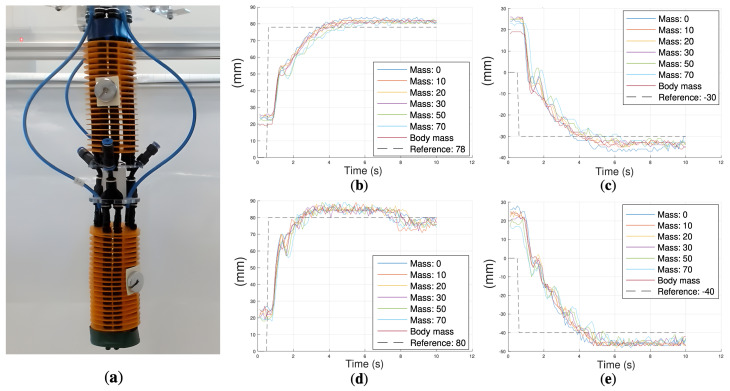
(**a**) In the vertical configuration, masses of 20 g are added to the lateral sides of the distal and proximal modules in an unbalanced distribution. Step responses of the robot in vertical configuration with the fractional-order controllers and for different masses: (**b**) $Ref_{x}=78$; (**c**) $Ref_{x}=-30$; (**d**) $Ref_{z}=80$; (**e**) $Ref_{z}=-40$.

**Figure 12 biomimetics-10-00502-f012:**
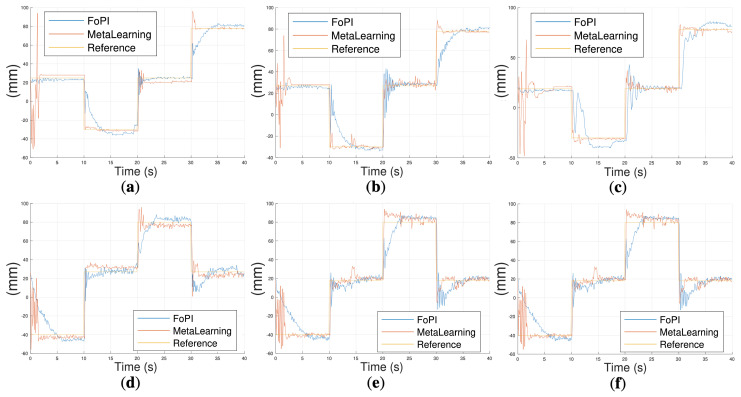
Vertical configuration comparison between FOPI and meta-learning approaches: on the X-axis (**a**) with no load at the robot’s tip, (**b**) with 50 g mass at the robot’s tip, and (**c**) with 20 g masses attached to the robot’s body; on the Z-axis (**d**) no load, (**e**) with 50 g mass, and (**f**) with 20 g masses attached to the robot’s body.

**Figure 13 biomimetics-10-00502-f013:**
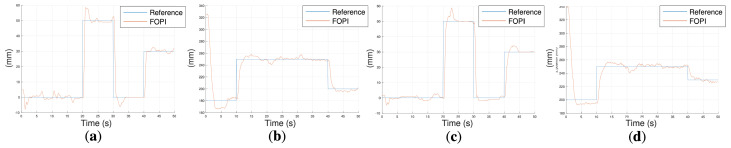
Horizontal configuration position control results with no load (**a**) on the X-axis and (**b**) on the Z-axis. Horizontal configuration position control results with 50 g mass at the robot’s tip (**c**) on the X-axis and (**d**) on the Z-axis.

**Figure 14 biomimetics-10-00502-f014:**
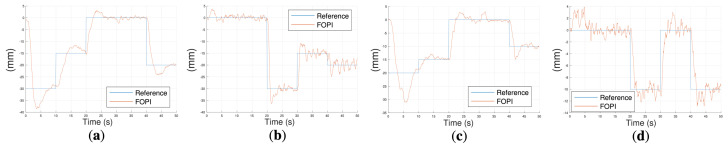
Horizontal configuration orientation control results with no load (**a**) on the transposed X’-axis and (**b**) on the transposed Z’-axis. Horizontal configuration orientation control results with 50 g mass at the robot’s tip (**c**) on the transposed X’-axis and (**d**) on the transposed Z’-axis.

**Figure 15 biomimetics-10-00502-f015:**
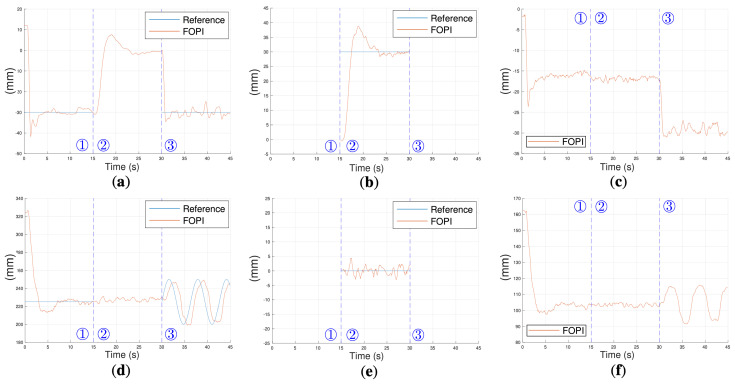
Horizontal configuration for the rubbing experiment: distal module position (**a**) on the X-axis and (**d**) on the Z-axis; distal module orientation (**b**) on the transposed X’-axis and (**e**) on the transposed Z’-axis; and proximal module position (**c**) on the X-axis and (**f**) on the Z-axis.

**Table 1 biomimetics-10-00502-t001:** Comparison between meta-learning-based control and the proposed IDMA+FOPI approach.

Approach	Control Structure	Orientation Handling	Initial Movement	Adaptability
Meta-learning	Data-driven with online adaptation and optimal control	No explicit orientation control	Large initial movements for exploration	High adaptability, high computational cost
IDMA + FOPI (proposed)	Decoupled model with FOPI control	Independent orientation control via distal module	Predictable response without initial exploration	Robust to loads and non-linearities, no training required

## Data Availability

The raw data supporting the conclusions of this article will be made available by the authors on request.

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
