# Peer review of "A Novel Enhanced Methodology for Position and Orientation Control of the I-SUPPORT Robot"

_biomimetics, 2025, doi:10.3390/biomimetics10080502_

Round 1
Reviewer 1 Report
Comments and Suggestions for Authors
Excellent presentation. It would be interesting to see how varying the orders of the controllers makes a difference.
Are the authors sure that the order of the loadings for the body are in the correct order in figure 12? Might the heavier load slow down the system in (c) vs. (b)?
Author Response
Manuscript ID: biomimetics-3775733
Title: A Novel Enhanced Methodology for Position and Orientation Control of the I-SUPPORT robot
Authors: Carlos Relaño *, Zhiqiang Tang, Cecilia Laschi, Concepción A. Monje
We would like to sincerely thank the reviewer for their time and valuable comments, which have helped us to improve the clarity and overall quality of the manuscript. We have carefully considered each suggestion and revised the manuscript accordingly. Below we provide a detailed, point-by-point response to the comments raised.
Comment 1: Excellent presentation. It would be interesting to see how varying the orders of the controllers makes a difference.
Response 1: Thank you for this valuable observation. The fractional order in the integral term of the FOPI controller directly affects the control system performance. In general terms, lower fractional order values increase low-frequency robustness and reduce overshoot sensitivity, but at the cost of slower steady-state convergence, whereas higher values provide an opposite performance.
A full parametric sweep of fractional orders was beyond the scope of this work. In our implementation, the fractional orders were specifically tuned to achieve the iso-damping property using the iso-m tuning method, as described in section 3.1.1, ensuring that the open-loop phase margin remains nearly constant for gain variations. This property leads to consistent transient response and robustness against parameter uncertainties, which is particularly relevant for pneumatic soft actuators that exhibit significant nonlinearities and variable stiffness.
Comment 2: Are the authors sure that the order of the loadings for the body are in the correct order in figure 12? Might the heavier load slow down the system in (c) vs. (b)?
Response 2: Thank you for pointing this out. We have reviewed the data and confirm that the figure order is correct. The additional mass in configuration (c) increases endpoint inertia, which leads to reduced high-frequency vibration levels but also causes a slightly higher overshoot due to the actuation dynamics.
Reviewer 2 Report
Comments and Suggestions for Authors
This paper presents a novel control methodology for the I-SUPPORT soft robot, enabling precise position and orientation control using fractional order controllers and kinematic decoupling. It introduces the Independent Distal Module Actuation (IDMA) method to decouple control of the proximal (position) and distal (orientation) modules. The approach is validated in both vertical and horizontal configurations, showing robust performance under varying loads. Compared to meta-learning-based control, it avoids large initial movements and offers better stability. The method also enables complex tasks like gentle rubbing during bathing, advancing soft robotics for assistive applications. This paper is well-structured and well-written. Before publication, there are some questions to be solved.
- The authors mentioned “The flexibility, adaptability, and safety features of soft robots have facilitated the exploration of novel possibilities in diverse fields, including healthcare, manufacturing, and space exploration.”, more state-of-the-art can be cited: DOI: 10.34133/cbsystems.0105; DOI: 10.34133/cbsystems.0192.
- Considering the gravitational effects are non-negligible, how to validate the constant curvature assumption?
- How to define the decoupling accuracy between cable and pneumatic actuation, and how to measure the error?
- In the horizontal configuration, the proximal module uses cable actuation, while the distal uses pneumatic, how this affect the results?
- What’s the maximum frequency that the proposed controller can track?
- The orientation control uses a straight-line reference, how about the noise of the measurement device?
- The transfer function identification assumed no delays, it seems this is not suitable for pneumatic actuation.
Author Response
Manuscript ID: biomimetics-3775733
Title: A Novel Enhanced Methodology for Position and Orientation Control of the I-SUPPORT robot
Authors: Carlos Relaño *, Zhiqiang Tang, Cecilia Laschi, Concepción A. Monje
We sincerely thank the reviewer for the thorough reading of our manuscript and the insightful comments, which have helped us to improve the quality and clarity of our work. We have carefully considered each point raised and revised the manuscript accordingly, as detailed below. These changes have been highlighted in red in the revised manuscript.
Comments 1: The authors mentioned “The flexibility, adaptability, and safety features of soft robots have facilitated the exploration of novel possibilities in diverse fields, including healthcare, manufacturing, and space exploration.”, more state-of-the-art can be cited: DOI: 10.34133/cbsystems.0105; DOI: 10.34133/cbsystems.0192.
Response 1: Thank you for this valuable suggestion. We have added the suggested references to enrich the state-of-the-art discussion on the applications of soft robotics (page 2, lines 36–39). The complete citations are now included in the bibliography references list, and the text has been modified as follows:
"Moreover, recent reviews highlight that progress toward intelligent soft robots depends on tightly integrated actuation-sensing and on soft, tissue‑compliant materials that enable safe, long‑term human interaction [9,10]." Where 9 is Zhou et al. (2024) and 10 is Wu et al. (2025).
Comment 2: Considering the gravitational effects are non-negligible, how to validate the constant curvature assumption?
Response 2: We appreciate your remark on this important point. We agree that the constant curvature assumption is an idealization and cannot be perfectly satisfied in real scenarios, particularly due to gravitational effects and load-dependent deformations. Instead of relying on a perfect constant curvature model, our approach explicitly addresses this modeling gap through a robust fractional-order control strategy, which is designed to handle uncertainties and deviations from the nominal constant curvature model.
The experimental results in both vertical and horizontal configurations demonstrate that, despite the modeling simplification, the proposed controller maintains accurate position and orientation tracking, validating the control-oriented nature of the approach. We have clarified this reasoning in the revised manuscript (page 22-23, lines 694-699):
"It is worth noting that, while the constant curvature assumption is widely adopted for its modeling simplicity, it is not strictly satisfied in real conditions due to gravity and load-dependent deformations. In this work, it is considered only as an initial approximation, with a robust fractional-order control strategy compensating for deviations from ideal curvature. Experimental validation in both vertical and horizontal configurations demonstrate accurate position and orientation tracking despite these modeling limitations."
Comment 3: How to define the decoupling accuracy between cable and pneumatic actuation, and how to measure the error?
Response 3: Thank you for this insightful question. In our framework, decoupling accuracy is conceptually defined as the extent to which actuation of one module influences only its intended controlled variables, without affecting the other module. Ideally, the proximal cable actuation affects only position, and the distal pneumatic actuation affects only orientation, with no cross-effects.
In real experiments, although relatively small, some coupling effects are present and can be measured experimentally by applying controlled inputs to one module while keeping the other at rest and recording the resulting motion of both modules. However, accurately modeling these effects is challenging due to the nonlinear nature of the soft robotic system.
This is precisely one of the points where the robustness of the proposed fractional-order control approach stands out: it achieves accurate position and orientation tracking even in the presence of these coupling effects, demonstrating resilience to modeling mismatches and actuation interactions.
Comment 4: In the horizontal configuration, the proximal module uses cable actuation, while the distal uses pneumatic, how this affect the results?
Response 4: Thank you for raising this point. The horizontal configuration introduces additional gravitational effects that mainly act on the distal module. By assigning cable actuation to the proximal module, we ensure a stiff and accurate positioning of the base structure, while the pneumatic actuation at the distal module provides compliant orientation control. This division of actuation tasks helps to mitigate the gravitational impact: the cable actuation resists sagging, while the pneumatic module accommodates orientation commands without inducing large coupling effects.
As a result, while the horizontal configuration exhibits slightly slower transient responses compared to the vertical one, the overall tracking accuracy remains within the range required for the intended assistive tasks.
Comment 5: What’s the maximum frequency that the proposed controller can track?
Response 5: Thank you for this question. The proposed control strategy is not fundamentally limited by a fixed maximum tracking frequency. Instead, the selected operating speed is determined primarily by the maximum allowable load, the morphology of the soft robot, and safety considerations for assistive tasks. As explained in the manuscript, we designed the controller for a predefined nominal operating speed, ensuring robust performance across different loading conditions relevant to bathing assistance.
If higher operating speeds were required, the controller could be re-tuned for those conditions without fundamental limitations from a control design perspective. However, the physical constraints of the soft robot -its compliant structure and pneumatic actuation- impose practical limits on how fast motions can be safely executed without compromising the integrity of the robot or user safety. For these reasons, we focused on reasonable speeds suitable for safe physical human–robot interaction, such as gentle back washing, rather than on pushing the frequency limit of the actuation system.
Comment 6: The orientation control uses a straight-line reference, how about the noise of the measurement device?
Response 6: Thank you for this relevant comment. Indeed, the orientation measurements obtained from the Intel RealSense depth camera are affected by sensor noise, introducing small fluctuations (~1 mm) into the control reference. As shown in Figure 15(b), during the second phase -when the orientation of the distal end is controlled- the desired orientation is achieved with very small error, indicating that the result is accurate. Although measurement noise inherently affects accuracy and control precision, the proposed fractional-order control approach demonstrated sufficient robustness to maintain stable and accurate orientation tracking despite these disturbances.
Nonetheless, we acknowledge this limitation, and we expect that further performance improvements could be achieved using sensors with higher precision or employing additional filtering strategies. This limitation has been explicitly mentioned in the revised manuscript (page 6, lines 189-191):
"The Intel RealSense depth camera used has a resolution of approximately 1 mm, and measurement noise can introduce small fluctuations. However, it is sufficiently accurate for the applications proposed in this work."
Comment 7: The transfer function identification assumed no delays, it seems this is not suitable for pneumatic actuation.
Response 7: Thank you for highlighting this important aspect. Indeed, pneumatic actuators typically introduce delays that can impact control performance. However, in our particular setup, the delays were experimentally observed to be negligible due to the small dimensions of the robot and the limited air volumes within its chambers, which remain partially inflated during operation. Under these conditions, the actuator responses exhibited minimal latency, validating our assumption of negligible delay in the identified transfer functions.
This assumption is further supported by the experimental results, as stable and accurate tracking was consistently demonstrated without noticeable delay-induced performance degradation. Nonetheless, we recognize that explicit delay modeling might be necessary for larger-scale pneumatic systems or faster operation scenarios, and we have explicitly noted this consideration in the revised manuscript (page 14, lines 458-464):
"Although pneumatic actuators typically exhibit delays, in our setup these were experimentally observed to be negligible due to the small dimensions of the robot and the limited air volumes of its chambers, which remain partially inflated during operation. Consequently, the identified transfer functions were modeled without delay, an assumption supported by the experimental results showing stable and accurate tracking without noticeable latency effects. Delay modeling may, however, be required in larger or faster pneumatic systems."
Reviewer 3 Report
Comments and Suggestions for Authors
This manuscript presents a technically solid and experimentally grounded control framework for the I-SUPPORT soft robot. The combination of fractional order control and independent distal module actuation (IDMA) enables effective decoupling of position and orientation control, and the experimental validation is clearly structured. The comparison with meta-learning-based control is insightful, and the implementation of fractional order controllers is well justified for a compliant robotic platform.
The manuscript is overall strong and clearly written, with a notable contribution to soft robot control architecture. I recommend minor revision, with a few comments and clarifications listed below.
1. Could the authors provide a table summarizing the differences between the proposed IDMA + FOPI approach and the previously applied meta-learning-based control? A comparison in terms of control structure, orientation handling, initial movement behavior, and adaptability would clarify the novelty of the contribution.
2. If the robot were scaled up in size or modified with more modules or heavier actuation structures, would the current control framework still apply? A brief discussion on generalizability to larger or structurally different I-SUPPORT platforms would be appreciated.
3. Could the authors briefly discuss what practical performance improvements were observed in the robot's behavior as a result of the proposed orientation control strategy? For instance, in real-world tasks such as the rubbing experiment, was there a measurable or observable benefit in terms of stability, consistency, or accuracy? A short application-level discussion of how the new control approach enhanced task performance would strengthen the impact of the results.
4. The authors mention that the robot’s resting position is not zero due to the curvature of its modules in the absence of actuation (Section 4.1.1, Figure 11(a)). Could the authors comment briefly on whether this initial curvature affects the control accuracy at the beginning of motion? It would be helpful to clarify if any compensation or adjustment was required in the control design.
5. Minor corrections:
"differents masses" → "different masses" (Section 4.1.1 header)
"staring at time 40 seconds" → "starting at time 40 seconds" (Section 4.2.1)
Author Response
Manuscript ID: biomimetics-3775733
Title: A Novel Enhanced Methodology for Position and Orientation Control of the I-SUPPORT robot
Authors: Carlos Relaño *, Zhiqiang Tang, Cecilia Laschi, Concepción A. Monje
We thank the reviewer for the positive evaluation of our work and for the constructive comments, which have helped us to improve the clarity and impact of the manuscript. We have carefully addressed all the points raised, as detailed below. These changes have been highlighted in red in the revised manuscript.
Comment 1: Could the authors provide a table summarizing the differences between the proposed IDMA + FOPI approach and the previously applied meta-learning-based control? A comparison in terms of control structure, orientation handling, initial movement behavior, and adaptability would clarify the novelty of the contribution.
Response 1: Thank you for this excellent suggestion. We have added a new table summarizing the main differences between the proposed IDMA + FOPI approach and the previously implemented meta-learning-based control. The table compares control structure, orientation handling, initial movement behavior, and adaptability. This new table has been included in Section 4.1.2. Comparison of FOPI and meta-learning approaches (page 20, lines 620-622) and Table1 of the revised manuscript:
"Table 1 summarizes the main differences between the previously applied meta-learning-based control and the proposed strategy in terms of control structure, orientation handling, initial movement behavior, and adaptability."
Comment 2: If the robot were scaled up in size or modified with more modules or heavier actuation structures, would the current control framework still apply? A brief discussion on generalizability to larger or structurally different I-SUPPORT platforms would be appreciated.
Response 2: Thank you for this insightful question regarding the generalizability of the proposed control framework. Conceptually, the Independent Distal Module Actuation (IDMA) approach combined with fractional-order control is modular by design and thus adaptable to different robot sizes or to configurations with additional modules. The decoupling strategy would still be applicable because it relies on module-level kinematic relationships and is not limited to a specific geometry.
Indeed, as noted in the manuscript, the methodology can be extrapolated to a soft robot with a greater number of modules, although this requires adding new angular references for intermediate modules and careful configuration parameter selection to avoid control redundancy and overlapping actions. Scaling up the robot or incorporating heavier modules introduces additional challenges at the hardware level, mainly due to increased inertial and gravitational loads, which may demand actuators with greater force output and improved sensing.
While the proposed control framework remains applicable, these scenarios would require re-tuning the fractional-order control parameters and potentially redesigning the actuation hardware to maintain the same level of performance. A brief discussion on these aspects has been included in the revised manuscript (page 23, lines 724-729):
"Furthermore, the modular nature of the proposed IDMA control framework allows its extension to I-SUPPORT configurations with additional modules or scaled-up geometries. While the decoupling strategy remains valid, increasing the number of modules or their size amplifies kinematic complexity and actuator loads, requiring re-tuning of fractional-order parameters and potentially more powerful actuation and improved sensing to maintain the same performance."
Comment 3: Could the authors briefly discuss what practical performance improvements were observed in the robot's behavior as a result of the proposed orientation control strategy? For instance, in real-world tasks such as the rubbing experiment, was there a measurable or observable benefit in terms of stability, consistency, or accuracy? A short application-level discussion of how the new control approach enhanced task performance would strengthen the impact of the results.
Response 3: Thank you for this opportunity to clarify the practical performance improvements derived from the proposed orientation control strategy. As shown in the results section, introducing explicit orientation control significantly improved endpoint stability and motion consistency, reducing transient oscillations and enhancing trajectory tracking during the gentle rubbing experiment. This produced smoother contact interaction, reduced slip during surface following, and improved repeatability of the task compared to previous approaches where orientation was not explicitly regulated.
Beyond the rubbing experiment, the capability to control both position and orientation increases the applicability of the I-SUPPORT platform to a broader range of tasks. Orientation control is particularly relevant in applications such as inspection and surface treatment, where precise alignment with complex geometries is required. Therefore, equipping the robot with orientation control not only improves performance in the demonstrated bathing scenario but also increases its potential for assistive and industrial applications that demand accurate and stable endpoint orientation.
A short discussion of these benefits has been added to the revised manuscript (page 23, lines 703-708):
"The explicit orientation control strategy improved endpoint stability and trajectory consistency, reducing transient oscillations and enabling smoother contact interaction during the rubbing task. This resulted in more repeatable performance compared to previous approaches without orientation regulation and expands the potential of the I-SUPPORT platform to tasks requiring precise alignment with surfaces or instruments."
Comment 4: The authors mention that the robot’s resting position is not zero due to the curvature of its modules in the absence of actuation (Section 4.1.1, Figure 11(a)). Could the authors comment briefly on whether this initial curvature affects the control accuracy at the beginning of motion? It would be helpful to clarify if any compensation or adjustment was required in the control design.
Response 4: We appreciate this observation. The initial curvature does not affect the controller’s ability to reach the desired positions because the control strategy regulates motion relative to the defined reference configuration. In other words, even though the resting position is not at zero, the robot can still achieve any commanded pose within its workspace without additional compensation. The robustness of the proposed fractional-order controller effectively compensated for this offset, making additional calibration unnecessary. We have clarified this point in the revised manuscript (page 18-19, lines 590-593):
"However, this initial curvature does not affect control accuracy, since the proposed control strategy regulates motion relative to the resting configuration, and the fractional-order controllers robustly handle this offset without requiring explicit calibration or compensation."
Comment 5: Minor corrections:
"differents masses" → "different masses" (Section 4.1.1 header)
"staring at time 40 seconds" → "starting at time 40 seconds" (Section 4.2.1)
Response 5: Thank you for noting these typographical errors. We have corrected them in the revised manuscript.
Round 2
Reviewer 2 Report
Comments and Suggestions for Authors
accept